# Influence of Food Safety Concerns and Satisfaction with Government Regulation on Organic Food Consumption of Chinese Urban Residents

**DOI:** 10.3390/foods11192965

**Published:** 2022-09-22

**Authors:** Duo Chai, Ting Meng, Dong Zhang

**Affiliations:** 1School of Government, Central University of Finance and Economics, Beijing 100081, China; 2Academy of Global Food Economics and Policy, Beijing Food Safety Policy and Strategy Research Base, College of Economics and Management, China Agricultural University, Beijing 100083, China; 3School of Public Administration and Policy, Renmin University of China, Beijing 100872, China

**Keywords:** organic food, food safety concern, satisfaction with government regulation, theory of planned behavior, willingness to consume, Chinese urban residents

## Abstract

In order to study the impact of food safety concerns and government regulation on Chinese urban residents’ organic food consumption willingness and behavior, an “online + offline” survey of 799 urban residents in Beijing has been conducted. Based on the theory of planned behavior, a structural equation model (SEM) was established and the government’s food production support regulation (GP) and sales guarantee regulation (GC) were incorporated separately into the SEM as moderator variables. The path influence coefficients of respondents’ food safety concerns, subjective norms, and perceived behavioral control on organic food consumption willingness were 0.065 (*p* < 0.05), 0.174 (*p* < 0.01) and 0.574 (*p* < 0.01), respectively. The influence of GP on organic food consumption willingness was 0.243 (*p* < 0.01), but its moderating effect on the promotion effect of food safety concerns and attitude to organic food consumption willingness was −0.001 (*p <* 0.01). The moderating effect of GC on the transformation from consumption willingness to behavior was 0.083 (*p <* 0.05). The results show that respondents attach the most importance to the comparison of costs and the benefits of organic food. Although the government’s food production regulation will weaken the driving effect of food safety concern and benefit perception of organic food consumption willingness, it will still promote organic food consumption willingness on the whole. The government’s supervision of food processing and sales is conducive to the occurrence of organic food consumption behavior.

## 1. Introduction

The influence mechanisms of organic food consumption intention and behavior have been studied for a long time [1,2]. Although the impact of organic food on human health remains to be determined, it helps to reduce food safety risks such as pesticide residue and excessive additives [3]. Promoting the cultivation and breeding of organic food can reduce the application of chemical fertilizers, pesticides, veterinary drugs and growth regulators [4], so as to control agricultural non-point source pollution and reduce accidental damage to the ecological chain caused by the chemical control of pests and diseases [3]. The processing, packaging, transportation and storage of organic food will also reduce the use of antibacterial and antiseptic agents, artificial pigments and other additives [5]. Publicizing and selling organic food also helps to enhance the social awareness of environmental protection [6]. Therefore, it is reasonable to study organic food consumption willingness and behavior from the perspective of the public’s awareness of food health and environmental protection, but this awareness is affected by complex factors such as personal conditions, prices, information and government regulation, etc., which is a proposition worth studying.

From 2001 to 2021, China’s per capita food output increased from 355 kg to 483 kg (data comes from the National Bureau of Statistics of China). At the same time, the excessive pursuit of food production and the rapid industrialization also caused serious water and soil pollution: in 2019, the amount of pesticide used in China was about 1.7737 million tons, accounting for 20% of the world (data comes from FAO), while farmland accounted for only 7%. In 2021, the amount of fertilizer applied to crops per hectare in China was 506.11 kg, 2.05 times that of Britain, and 3.69 times that of the United States (China Agriculture Green Development Report 2021 by the Chinese Academy of Agricultural Sciences). According to the survey conducted by the Ministry of Agriculture and Rural Affairs of China in 2020, 30% of the sampled soils were continuously polluted. In food processing, problems such as food hygiene, abuse of additives, and improper packaging and transportation are also serious [7]. In this context, Chinese residents experience high levels of anxiety about food quality safety. According to the Baidu Index, there were more than 1.3 million pieces of online news with the theme of food safety in China in 2021, while the average number of daily searches for keywords related to “organic food” exceeding 80,000 (there are also similar terms such as green food, pollution-free food, and pure natural food). After the “melamine milk powder” incident and other food safety crises, Chinese residents’ consumption of imported organic food increased rapidly.

According to the statistics of the State Administration for Market Regulation of China and the Foresight Industry Research Institute, the scale of the organic food industry in China increased from CNY 27.98 billion to CNY 67.821 billion from 2013 to 2019, and is expected to reach CNY 98.984 billion in 2023. Promoting the organic food market is significant for agricultural upgrading, increasing farmers’ incomes, and protecting the ecology of China’s agriculture areas. At present, scholars have conducted some research on the influence of residents’ personal characteristics and environmental awareness on organic food consumption [8,9,10]. Although it can be observed that food safety concerns and the mistrust of government regulations may be factors that encourage Chinese residents to buy organic foods, the mechanism has not been fully addressed, and the role of government regulation is unclear. Understanding the organic food consumption psychology of Chinese residents is helpful for government policies and for enterprises to develop in the fast-growing market.

In order to study the impact of food safety concerns on Chinese residents’ organic food consumption willingness and behavior, and analyze the role of the government’s food production and sales supervision and support, an “online + offline” survey of 799 respondents in Beijing has been conducted in this study. The marginal contributions include: (1) Explaining the driving mechanism behind residents’ pursuit of healthy food and the demand for organic food due to their own food safety concerns, environmental pressure, and “cost–benefit” perception, and comparing the impact of the three. (2) The moderating effect of satisfaction with government regulation on the entire chain of food production and consumption is divided into two parts: “the impact on residents’ food safety concerns” and “the impact on the conversion of consumption willingness into behavior”, improving the model of theory of planned behavior. (3) The concepts of “eco-tourism to green and organic farming villages” and “participation in food safety supervision” are also incorporated into the constituent factors of consumption willingness and behavior, extending the concept of organic food consumption.

## 2. Theoretical Background

### 2.1. Methodology

The theory of planned behavior (TPB) believes that behavior intention is jointly determined by attitude toward the behavior (attitude), subjective norms, and perceived behavior control [11]. Attitude refers to an individual’s evaluation of a behavior, which is jointly determined by the degree of cognition, preference, recognition ability, etc., of the behavior [12]. Subjective norms refer to the external pressure faced by individuals, including the influence of relatives, friends, neighbors, etc., on one’s thoughts, and social values such as morality, public opinion, laws and regulations [13]. Perceived behavior control refers to an individual’s judgment on the difficulty of behavior and the ability to control oneself [14].

TPB is widely used in the study of the impact of factors such as transgenic technology [15], brands [16], production areas [17], taste and quality [18], ecological environment [19,20], etc., on food consumption. In the study of organic food consumption, the role of TPB in studying the effects of personal characteristics, environmental awareness, food prices and other factors on the consumption willingness and behavior towards organic food has been confirmed [16,21].

However, many scholars notice that the willingness to consume under the TPB framework does not always lead to consumption behavior [22,23]. The reason for the deviation may be that the willingness to consume and the conversion process from willingness to behavior are interfered with by external factors that are not included in the TPB model [23]. Evidence from China shows that government publicity and supervision have a key impact on consumer demand [24]. Ma et al. [25] used game theory to analyze the role of government supervision in promoting the trust in and purchase demand of organic food; Wu et al. [26] verified that policies can change public perception and regulate consumption. These studies support the idea of introducing residents’ perception of government regulation to improve the TPB model [24].

### 2.2. Food Safety Concern and Food Choice

Psychologically, the influence of consumers’ food safety concern on purchasing organic food can be explained as the change in food consumption behavior caused by “the nervousness about food safety or conformity, the demand for one’s own health, and the dissatisfaction with the social environment” [14]. This is manifested in the pursuit of agricultural products with a more primitive production mode (less human intervention), and the willingness to pay a higher price for them [27]. It is believed that the causes of concern include subjective factors and external shocks [13]. In food safety issues, 28 individual subjective factors mainly refer to their own conditions and individual controllable factors. Jackson et al. [28] and Gerbens et al. [29] believed that with an increase in income and age, people pay more attention to food safety. Ansari et al. [27] found that home address and education level would significantly affect consumers’ concerns about food quality and safety. Xu et al. [30] believed that families with elderly members and children would be more sensitive to the quality of food and vegetables. External shocks mainly include serious food safety cases or food technology innovation. Pivato et al. [31] believed that the trust crisis caused by food safety scandals would prompt consumers to substitute organic food and other functional food for ordinary food. For example, the clenbuterol pork incident in China has significantly increased consumers’ willingness to buy organic pork [32]. According to Mutiri et al. [33], consumers’ sense of benefit that “organic food can resist disease and improve health” would motivate them to replace other food with organic food. Dai et al. [34] proposed earlier the logic of influence of Chinese consumers’ food safety perception on their willingness to consume organic food.

At present, the demand of Chinese residents for “eating nutritiously and healthily” has increased rapidly, and people are willing to pay high prices for “health concept” food [35]. Although the correlation between the food safety concern of Chinese residents and the increasing demand for organic food has been observed, we do not know how food safety concerns affect residents’ consumption psychology regarding organic food and from what aspects. Which factors have the greatest impact? How does consumption willingness turn into behavior? What role does government regulation play? Systematic research on these questions is still lacking [29].

Referring to the views of previous scholars, we can make the following assumptions: Firstly, exploring the formation mechanism of organic food consumption intention should start from understanding people’s understanding of food safety. At the same time, the urgency of the problem is also an important factor affecting attitude [27]. Residents’ understanding and the degree of concern (views and positions, including the need to pursue health) of the current situation of their own food safety, as well as their feelings about the environment of food production and sales they are exposed to (urgency) jointly determine the “subjective attitude” of residents towards food safety, which may be the initial motivation for their willingness to buy organic food [34]. We hereby hypothesize:

**Hypothesis** **1** **(H1a).***The deepening of Chinese residents’ awareness of and concern over food safety status will increase their willingness to buy organic food*.

Secondly, judgments, recognition, persuasion and the recommendations of relatives and friends may also form a psychological implication of “organic food with high value” for residents [36] which can significantly affect personal feelings about well-being, and “outsiders” such as colleagues from work have an even stronger demonstration effect [37]. In addition to regulations and ethics, news about food safety issues and the social environment of public opinions may enhance residents’ concern that “ordinary food cannot guarantee health” [37,38], enhancing their willingness to consume organic food. We hereby hypothesize:

**Hypothesis** **1** **(H1b).***The public opinion environment of food safety in society and the persuasion of others will enhance the willingness of Chinese residents to buy organic food*.

Thirdly, residents’ food consumption is also limited by various internal and external factors which can be mainly attributed to the perceived cost and perceived benefit of purchasing organic food. Under the existing agricultural production conditions and the scientific and technological levels in China, in order to give priority to ensuring “sufficiency”, resources such as land for organic food production may be limited, and cannot meet the needs of all residents to eat organic food [39]. Organic food is of higher production costs and selling prices, and is sold via relatively few channels. When considering purchasing organic food, residents need to think deeply and compare the “comprehensive cost (economic cost, energy cost, etc.)” and “health benefits (reducing diseases, increasing nutrition, bringing pleasure, etc.)”. In terms of research methods, the influence of perceived costs and perceived benefits can be combined into one structural variable after isotropic treatment [33] to be studied together, so as to measure the “perception of net benefits” of respondents after the consideration of costs. We hereby hypothesize:

**Hypothesis** **1** **(H1c).***After considering the costs of purchasing organic food, the higher the residents’ comprehensive sense of net benefits, the stronger their willingness will be to buy organic food*.

The willingness to consume is the possibility or tendency to generate consumption behavior [11], although deviation does exist [22]. Based on the analysis mentioned above, the behavior of Chinese residents in consuming organic food is a decision made from personal needs after considering others’ opinions and comparing costs and benefits. We hereby hypothesize:

**Hypothesis** **1** **(H1d).***Chinese residents’ willingness to consume organic food after the comprehensive consideration of various factors will form the consumption behavior of organic food*.

Furthermore, China is still in the process of increasing food demand, with limited purchasing power and high price sensitivity [30]. In 2021, the income level of Chinese residents was USD 5192, which is far behind that of developed countries. It is difficult for Chinese residents to completely replace ordinary food with organic food regardless of costs. It is found that, at present, Chinese food producers still attach importance to price competition, and in most cases, the quality competition will have lower benefits and higher risks [13]. This reflects that the influence of price on consumers’ choices of food may be greater than that of taste, appearance, publicity and other people’s opinions. We may speculate that under the restriction of the current income level, Chinese residents may pay more attention to cost–benefit comparison when considering buying organic food. We hereby hypothesize:

**Hypothesis** **1** **(H1e).***The influence of “cost–benefit” comparison on Chinese residents’ willingness to buy organic food is greater than that of food safety concern and that of the surrounding environment and others*.

### 2.3. Satisfaction with Government Regulation and Food Choice

Other than external persuasion, the impact of external interventions on individual psychology has been rarely considered in the traditional TPB model, which may lead to deviations between feelings and willingness, and willingness and behaviors [23]. Existing studies believe that the government’s financial and technical support for agricultural production [40] and the protection of the agricultural and rural environment can improve the enthusiasm of food producers to make rational use of means of production to produce healthy agricultural products, and also help alleviate residents’ anxiety about food safety.

The Chinese government plays a leading role in social governance and market supervision [41]. In the path of willingness formation and conversion into behavior in the traditional theory of planned behavior, the work of the Chinese government may have an impact on residents’ consumption psychology regarding organic food in two aspects.

First of all, the government can intervene in food production through agricultural subsidies and preferential policies, as well as by controlling the use of livestock and poultry drugs, fertilizers and pesticides, maintaining and building the farmland ecological environment and rural living environment [10], controlling agricultural non-point source pollution [33], and supporting the research, development and application of the science and technology of green agriculture [20], which is called the “government’s food production support work”. If residents are satisfied with the government’s support and supervision of agricultural production, their trust in the production process and quality of organic food may be enhanced; that is, they are willing to believe that agricultural producers, production resources, and the environment have the willingness and conditions to produce qualified organic food which may increase the willingness to consume organic food. We hereby hypothesize:

**Hypothesis** **2** **(H2a).***In general, the higher the residents’ satisfaction with the government’s food production support work, the higher the willingness to consume organic food*.

However, the higher the residents’ satisfaction with the government’s food production support work, the lower the food safety concern may be, and the weaker the benefit perception that “the utility of organic food is higher than that of ordinary food”. Thus, this may weaken the driving effect of food safety concern (attitude) and the perception of the net benefit of organic food (perceived behavior control) on the consumption willingness of organic food. We hereby hypothesize:

**Hypothesis** **2** **(H2b).***Higher satisfaction of residents with the government’s food production support work may weaken the driving effect of food safety concern on the willingness to consume organic food*.

**Hypothesis** **2** **(H2c).***Higher satisfaction of residents with the government’s food production support work may weaken the driving effect of the benefit perception of organic food on the willingness to consume organic food*.

Secondly, the government can promote residents’ willingness to trust and buy organic food by improving food laws and regulations, punishing food safety issues [25], providing authoritative information and certification, standardizing food packaging information, supervising food processing, transportation and storage, carrying out food safety publicity and education, and supporting food safety public welfare and charities [35], which can be called the “government’s food consumption guarantee work”. If residents have a high evaluation of the government’s consumption guarantee work, it is reasonable to believe that residents have fewer concerns about consuming organic food, and their willingness to consume will more likely be transformed into consumption behavior. We hereby hypothesize:

**Hypothesis** **2** **(H2d).***Satisfaction with the government’s food consumption guarantee work will promote consumers to purchase organic food*.

**Hypothesis** **2** **(H2e).***Satisfaction with the government’s food consumption guarantee work will prompt consumers to convert their consumption willingness of organic food into behavior*.

### 2.4. Personal Conditions and Food Choices

In view of the characteristics of organic food with reference to other studies, this paper focuses on analyzing the influence of three types of personal characteristics on the sales of organic food: First, the price of organic food is relatively high. The higher the income of residents, the stronger the willingness to buy organic food may be [9]. Second, the health status of the elderly and children is generally a major concern of Chinese families, so the number of elderly members and children in the family will also affect the willingness to consume organic food [42]. Third, residents’ chronic diseases are also strongly related to food choices. It can be speculated that chronic diseases suffered by residents themselves and their families will also stimulate a willingness to buy organic food [9]. We hereby hypothesize:

**Hypothesis** **3** **(H3a).***The higher the residents’ income, the stronger their willingness to consume organic food*.

**Hypothesis** **3** **(H3b).***The more elderly and children in the family, the stronger the willingness to consume organic food*.

**Hypothesis** **3** **(H3c).***The more severe the chronic diseases suffered by individuals and family members, the stronger the willingness to consume organic food*.

## 3. Materials and Methods

### 3.1. Participants

The purpose of this study is to analyze the role of food safety concerns and satisfaction with the government among Chinese urban residents in the formation of consumption willingness and behavior regarding organic food. Respondents need to have knowledge and consumption experience of organic food and have feelings about food safety issues and government regulation. China has a vast territory with huge gaps in regional development. The consumption awareness and sales channels of organic food have not yet been popularized nationwide. Relatively speaking, economically developed cities such as Beijing and Shanghai have a huge demand for food but a low self-sufficiency rate of local food. It is difficult for residents to obtain organic food frequently from relatives and friends living locally or in the countryside. The per capita income level is high, but the workload is heavy, the demand for health is relatively strong, and the logistics conditions and online and offline sales channels of organic food are developed. Therefore, these places are regarded by the Chinese government as priority areas to promote organic foods, and are also the markets that domestic and international organic food manufacturers are focused on developing at present. At the same time, compared with the south (traditionally, the Qinling Mountains and Huaihe River in China are the dividing line between the north and the south), the natural conditions in northern China are poor, the water quality and air pollution problems are serious, and the residents’ food safety concern is more obvious. Beijing is located in the political center with developed information transmission. Residents have a high degree of understanding of government regulations. The proportion of migrant residents in the permanent resident population reaches 40%, leading to the diverse ethnic, religious and native place structure of the population. Therefore, the selection of Beijing residents as the respondents in this paper is in line with our research needs. It is representative of the national organic food market, and has predictive value for other regions in the future.

On this basis, this study sets two screening questions in the questionnaire: “Do you often shop or cook?” and “Have you ever been exposed to organic food?” Samples that choose “No” for any question will be excluded, because the respondents who do not often buy ingredients or cook lack the perception of organic food consumption, and the respondents who have not been exposed to organic food have difficulty making judgments on the value and cost of organic food, which will affect the accuracy and effectiveness of the results. In addition, whether the respondents answer seriously is checked through three instructed items as well as the duration of questionnaire filling.

### 3.2. Questionnaire Design

The questionnaire was compiled in Chinese. During the design process, 25 Beijing citizens were invited to complete two rounds of tests. The questions were checked for logical structure, omissions or repetitions, ambiguity of expression, etc., and some were rephrased or slightly revised. The questionnaire consisted of 7 parts. Except for alternative questions and instructed items, 34 core questions were finally included in the model analysis (see Table A1 in Appendix A for details).

Attitude of food safety concern: The respondents were mainly asked about their understanding, concerns and perceptions. First of all, after listing common food safety problems, we asked the respondents, “Do you think the food safety problems in daily diet are serious?” The answers were set with the adoption of the 5-point Likert scale, from (1) not serious to (5) very serious. Then, we asked the respondents, “Do you think there is sufficient supply of safe food in the market?” and “Do you believe in the safety labeling and publicity on food packaging?” Referring to the practice of Liu et al. [20], we asked the respondents, “Do you prefer food produced in areas with a good ecological environment?” so as to identify “whether the respondents are aware that the food may be unsafe, how worried they are about food insecurity, and whether they are willing to specially select the food that may be safe”.

Subjective norms of food safety anxiety: Referring to the ideas of Düsing et al. [13] and Koklic et al. [39], in terms of descriptive norms and the opinions of the respondents’ colleagues, friends, neighbors and other people around them, we asked about “the frequency of talking about food safety issues” and “the frequency of recommending organic food”. The answers were set within the 5-point Likert scale. In terms of injunctive norms, since the consumption of organic food is not subject to legal requirements and moral constraints, the “frequency of seeing news about food safety issues” was taken as a question with reference to the research by Kuo et al. [38].

Perceived behavior control: According to the foregoing discussion, the survey was carried out from two aspects: perceived cost and perceived benefit. Questions included “Do you think organic food is expensive compared to your income?” “Do you think organic food can reduce disease?” and “Do you think buying and eating organic food can make families happy and harmonious?” The answers were set with the 5-point Likert scale.

Satisfaction with government regulation: In terms of the government’s food production support work, the respondents were mainly asked about their satisfaction with the government’s supervision of the use of livestock and poultry drugs, fertilizers and pesticides; the protection of farmland and the rural environment; support for agricultural science and technology research and development; and agricultural subsidies. A total of 6 questions were set. In terms of government’s food consumption guarantee work, the respondents were mainly asked about their satisfaction with government food safety certification, the standardization of food packaging, the supervision of processing and transportation, the improvement of food safety laws, the implementation of food safety education and publicity, and support for the public welfare undertakings of food safety. A total of 7 questions were set. Answers were set using the 5-point Likert scale ranging from (1) very dissatisfied to (5) very satisfied.

Personal characteristics: In addition to gender, age, etc., the information surveyed mainly included the respondents’ income, the number of elderly people and children living together in the family for a long time, and the number of chronic diseases from which the respondents themselves and their family members are suffering. According to the common situations of Chinese social and family settings, the 5-segment method was used to set the question options.

Willingness to consume organic food: With reference to mature practices, the respondents were asked about their “willingness to buy organic food at current prices”, “willingness to recommend organic food to others” and “willingness to buy organic food as a gift to others”. In addition, recent studies have revealed that residents could support the development of the organic food market not only through purchases, but also by travelling to ecological rural areas, eating organic foods on site, enjoying the rural agricultural organic production environment, or participating in public welfare undertakings of food safety. On this basis, this paper expands the connotation of “willingness to consume” organic food, and includes “willingness to visit rural ecotourism sites’” and “willingness to appeal, supervise and report food safety issues” in the questionnaire.

Consumption behavior of organic food: Residents’ actual consumption behavior relating to organic food was measured with questions designed from three aspects: the monthly consumption of organic food, the proportion of organic food consumption of overall household food consumption, and the frequency of rural tourism. Since public participation in public welfare undertakings of food safety in China is still in its infancy, there were very few actual participants, and the answers were quite distinct, so they are not included in the measurement of behavior. Affected by COVID-19, this article extended its research time to 1–3 years.

### 3.3. Procedure

(1)Data Collection

Questionnaire collection started on 5 January 2022 and ended on 5 March 2022. There were two ways of distributing the questionnaires: random interviews on the streets and online surveys. The former were mainly conducted at the entrances of supermarkets and residential quarters in Beijing. Xicheng District and Haidian District in the central urban area of Beijing, Changping District and Shunyi District in the suburbs, and Tongzhou District in the outer suburbs were selected for the surveys to enhance spatial representation. A total of 15 supermarkets and 20 residential districts were involved, and a total of 550 questionnaires were collected. The online survey was conducted with questionnaires mainly being distributed through Credamo, the well-known online questionnaire platforms in China, as well as Questionnaire Star. A total of 732 questionnaires were collected. In the survey, the respondents’ right to informed consent was fully guaranteed, and the responses were completely voluntary and anonymous.

(2)Samples

After the questionnaires were collected, a total of 204 samples who did not often shop, cook or had not been exposed to organic foods were deleted. Furthermore, 32 samples with wrong answers to “instructed items” were deleted. Due to the large number of questions in the questionnaire, 53 online survey samples with an answering time of less than 300 s were excluded according to the feedback from the previous 25 test takers. At last, at the significance level of 17 degrees of freedom and *p* < 0.01 [43], outlier questionnaires were screened by the Mahalanobis distance test, and 799 questionnaires were finally obtained. In the subscales of behavior attitude, subjective norms, perceived behavior control, willingness to consume, and consumption behavior, the Z-scores were all greater than 1.96, indicating that there were no outliers in the single variable [44], and all 799 questionnaires were valid. At last, Cronbach’s alpha coefficients of all latent variables measured were above 0.7, indicating that the internal consistency and reliability of the questionnaires were acceptable [15].

(3)Model Analysis

Based on the theory of planned behavior, the measurement model and structural model of organic food consumption are constructed in this study through structural equation modeling (SEM). On this basis, Bartlett’s test is performed and KMO value is measured to test if the sample data are suitable for factor analysis, and the composite reliability (CR) index and average variance extracted (AVE) are introduced to test the reliability and validity of the model. A set of goodness of fit indicators, namely, Chi-square/df GFI, AGFI, and RMSEA, are also introduced to further demonstrate the validity of SEM. Meanwhile, the moderator variable of the government is added to investigate the moderating effect of government regulation on residents’ willingness and behavior to consume organic food. In behavioral science, the research data are usually not normally distributed, and the maximum likelihood estimate may be deviated. Therefore, the generalized least squares estimation is adopted as the estimation method in this study to eliminate the influence of data distribution on the estimation results. In the verification of the government’s moderating effect, the paired product method is used to set the interaction terms. This has the advantage that it can not only use all indicators as much as possible to make full use of information, but it can also prevent the multicollinearity problem caused by repeated use [45]. The specific steps are: (a) Centralize all the indicators. (b) According to the size of item loading, the indicators in the two latent variables are matched and multiplied according to the principle of “large with large, small with small”. If the number of indicators is not equal, it is necessary to merge or delete redundant items. (c) Build a model without mean structure using an unconstrained method [46]. SEM analysis is achieved by Mplus 8.3.

Meanwhile, this study controls the personal characteristics that may affect the willingness to consume organic food, including gender, age, education background, monthly income level, the number of elderly and children in the family, the number of chronic diseases, etc. The moderator variable “government’s food production support work” has a moderating effect on the paths in which behavior attitude and perceived behavior control affect consumption willingness, and the moderator variable “government’s food consumption guarantee work” has a moderating effect on the path of consumption willingness, affecting consumption behavior (Figure 1).

## 4. Results

### 4.1. Participants

As shown in Table 1, all 799 respondents are between the ages of 18 and 75, and there are fewer samples under the age of 18 and over the age of 60, which is in line with the characteristics that food is mainly purchased and cooked by young and middle-aged people in Chinese households. There are 297 males (37.17%) and 502 females (62.83%), which is basically in line with the characteristics that in Chinese households, food is mainly purchased and cooked by women. Since the respondents do not include minors, the educational structure of the sample basically conforms to that of residents in major cities in China. The income characteristics of the samples are generally in line with the normal distribution, which is also similar to the distribution characteristics of citizens’ income published by the Beijing Municipal Bureau of Statistics in June 2021. It indicates that the samples show appropriate representativeness.

### 4.2. The Measurement Models

The item loading, reliability and validity index of the measured variables are shown in Table 2. The model passed the Bartlett’s test with a significance level of 5%, and the KMO value was 0.933, indicating that the sample data were suitable for factor analysis. The results of the confirmatory factor analysis showed that all item loadings of the basic model were greater than 0.6, exceeding the threshold of 0.5 [44], which indicates that all questions were valid and could effectively reflect the measured model variables.

Next, the convergent validity of each latent variable was calculated to ensure the consistency and reliability of the internal structure, including CR and AVE [43]. The results showed that CR in the basic model was greater than the threshold of 0.7 [44], and AVE was greater than the threshold of 0.5 [43], indicating that the internal consistency of the latent variables was completely acceptable.

According to the Fornell–Larcker criterion, if the square root value of AVE of each latent variable is greater than the correlation coefficient with other latent variables, it indicates that it has good discriminant validity [36]. In Table 3, the diagonal data are the square roots of the AVE of the latent variables, and other data are the correlation coefficients between the latent variables. The square root of the AVE of each latent variable is higher than the correlation coefficient with any other latent variable, which conforms to the Fornell–Larcker criterion, showing good discriminant validity. In addition, Table A2 in Appendix B shows the item loading of latent variables to which all indicators belong as well as the cross-item loading with other latent variables. The results show that all off-diagonal item loading values are lower than the diagonal values, which also indicates that the measurement model has good discrimination.

### 4.3. The Structural Model

The multicollinearity test results of the model are shown in Table 4. The VIFs of all endogenous structural variables are below 2.1, which is less than the threshold of 5, indicating that there is no multicollinearity problem in the structural model. The coefficients of determination R^2^ of consumption willingness and consumption behavior are 0.62 and 0.43, respectively. In the study of consumer behavior, these values indicate that the research results have considerable substantive explanatory power [44]. In addition, the χ^2^ statistics of the structural model are significant (chi-square = 3242, *p* < 0.000, df = 902). Chi-square/df = 3.59, which is less than the criteria of 5. GFI = 0.921 and AGFI = 0.900; both are greater than 0.9. RMSEA = 0.057, which is less than 0.06. Based on the above-mentioned results, it can be considered that the model is of a good fitting effect.

### 4.4. Path Coefficients Result

Table 5 and Figure 2 show the estimation results of the path coefficients. The behavior attitude measured in this study is also known as food safety concern (β = 0.065); subjective norms are the opinions of others (β = 0.174); and perceived behavior control is residents’ perception of the net benefit of organic food after considering costs (β = 0.574). These factors have a positive impact on the willingness to consume organic food, which is statistically significant, verifying the hypotheses H1a, H1b and H1c. Figure 3 also shows the results of the responses for the alternative questions of the questionnaire, showing that the food safety concerns of Chinese residents are at a high level.

The willingness to consume has a high positive impact on consumption behavior (β = 0.578), indicating that the behavior of residents to consume organic food is formed by the willingness after considering various factors, which verifies H1d. It can be seen that the influence coefficient (0.574) of perceived behavior control, that is, the perception of net benefit, is much higher than that of subjective norms (0.174) and behavior attitude (0.065), which may indicate that under the current income level, Chinese urban residents will pay more attention to the cost–benefit comparison when purchasing organic food. They may be more sensitive to food prices, since they still doubt the benefits of the organic food, according to the answers to alternative questions in the questionnaire (shown in Figure 4). At the same time, Chinese urban residents generally attach importance to their relationship with surrounding people and have conformity [47], and pay high attention to social events. The influence of public opinion and surrounding people also has a significant impact on their feelings of food safety and willingness to buy organic food. In contrast, residents’ own attitudes towards food safety have less influence on their willingness to consume organic food, which is also in line with Chinese people’s characteristics of being budget conscious, being concerned about “face-saving”, and having high tolerance for food. The internal concern for their own food nutrition and health is gradually growing. The above results verify the hypothesis H1d.

### 4.5. Moderating Effect of Satisfaction with Government Regulation

In order to explore the moderating effect of residents’ satisfaction with government regulation on the formation of consumption willingness of organic food and the transformation of willingness into behavior, two moderator variables, i.e., the satisfaction with the government’s food production support work (GP) and with the government’s food consumption guarantee work (GC), are set according to the derivation of 2.3 in this paper. GP mainly regulates the influence paths of behavior attitude and perceived behavior control on consumption willingness, and GC regulates the influence path of consumption willingness on consumption behavior. All the answers related of participants are shown in Figure 5.

Table 5 shows the interesting results: in terms of overall effect, residents’ satisfaction with government’s food production support work has a positive impact on the willingness to consume organic food (β = 0.243), which verifies the hypothesis H2a. However, from the perspective of moderating effect, the satisfaction with government regulation shows a negative moderating effect (β = −0.100) on the influence path of residents’ subjective attitudes towards food safety concern on their willingness to consume organic food, indicating that the higher the residents’ satisfaction with the government’s food production support work, the lower the food safety concern may be, which has a certain inhibitory effect on the transmission path of “attitude—organic food consumption willingness”, thus verifying hypothesis H2b. However, the moderating effect of residents’ satisfaction with the government’s food production support work on the transmission path of “perceived behavior control—consumption willingness” is not significant, and H2c has not been verified.

The influence coefficient of satisfaction with the government’s food consumption guarantee work on consumption behavior (β = 0.172) is significantly positive. It also has a significant positive moderating effect on the transmission path of “consumption willingness—consumption behavior” (β = 0.083). H2d and H2e are verified.

### 4.6. Influence Analysis of Control Variables

There have been many studies on the influence of gender, age and education on food consumption. These are not the focus of this paper, and the results are not significant. Income level has a positive impact on the willingness to consume organic food (β = 0.106). The respondents with more chronic diseases are more willing to buy organic food (β = 0.073). The number of elderly family members and children living together also has a positive effect on the willingness to consume organic food, but the influence is weak (β = 0.007). To sum up, the hypotheses H3a, H3b and H3c are verified.

### 4.7. Hypotheses Tests

Table 6 gives an overview of the tested and confirmed hypotheses.

## 5. Discussion

Because of China’s special food supply situation and its residents’ consumption habits, the results of this study have obvious Chinese characteristics, and also have similarities with other countries.

### 5.1. The Particularity of the Influence of Chinese Residents’ Food Safety Concerns on Organic Food Consumption

It has been proven that the demand for food health is the cause of organic food consumption intention and behavior. However, there is a big difference between “hope food is healthier” and “feel food is unsafe”. The survey conducted by Guiné et al. [48] on consumers in Turkey and Portugal proves that in the context of COVID-19, consumers’ concern for health and environmental impact are the most important motivation for organic food consumption. The study of Śmiglak-Krajewska et al. [49] in Poland also confirms that the COVID-19 epidemic has strengthened consumers’ preference for health value when choosing food. The logic of these studies is that the COVID-19 epidemic has exacerbated people’s health concerns; in order to be healthier, residents will buy healthier food. The underlying premise here is that the residents of these countries fully understand and are willing to believe the advantages of organic food over traditional food [48].

Unlike European countries, China is at the initial stage of food consumption demand upgrading. Gong [47] believe that incidents such as the “melamine milk powder” and “gutter oil” events that once caused a sensation in China are one of the main reasons for residents’ food safety concerns, and are also the reason why more and more residents are willing to pay high prices for “food declared to be healthy”. In this study, when answering the alternative questions of the questionnaire “Do you think the food currently sold in markets and supermarkets is safe?” and “Do you think the food currently sold online is safe?”, 60% and 48% of the respondents chose “Less safe” or “Unsafe” (Figure 4), respectively, which directly indicates that the respondents are generally worried about food quality safety. In terms of attitude measurement, the respondents were asked gradually about their psychological feelings of “whether they understand that food may be unsafe, how worried they are about food insecurity, and whether they are willing to specially select food that may be safe”. The questions on subjective norms actually measured the impact of external pressure and suggestions on residents’ food safety concerns. The questions on perceived behavior control actually measured the residents’ comparison of comprehensive cost with the “safer benefits than ordinary food” of organic food. These questions reflect different aspects of food safety concerns.

Hypotheses H1a~H1d have been confirmed. The possible logic is that due to long-term problems such as pesticide, fertilizer abuse and soil pollution, residents are worried about food quality safety, so they will try to buy organic food that “claims that the production process is healthier but the price is more expensive”. This is a kind of “adverse selection”, because residents are willing to pay more for organic food just to avoid becoming ill.

Furthermore, due to the observation that the price of organic food in China is relatively high and that society has different opinions on the effects of organic food, this study uses “cost–benefit” comparison to reflect perceived behavior control in terms of planned behavior theory, and finds some new and interesting phenomena: the comparison of the costs and benefits of organic food by Chinese residents has a far greater impact on the consumption willingness of organic food than the attitude and subjective norms of food safety concerns. In contrast, Teixeira et al. [50] found in their study on Portugal that consumers’ perception of health problems and organic food quality is the main factor affecting organic food consumption intention, but did not clearly indicate the restrictive effect of cost factors. Guiné et al. [48] and Śmiglak-Krajewska et al. [49] also believe that the demand for health benefits is the main factor that promotes the consumption of organic food, and that residents will buy organic food at an affordable price. According to Zámková et al.’s research [51] in the Czech Republic, with an increase in income, the most important factor that consumers pay attention to when buying organic food will gradually change from “price” to “quality”. The results of this study show that, at the current income level, although Chinese residents hope to avoid food safety problems through organic food consumption, they cannot fully trust the efficacy of organic food and have doubts about whether the price of organic food is inflated. Compared with uncertain health benefits, doubts about the “true value of organic food” and “realistic cost-effectiveness” are the main factors affecting the willingness to buy organic food. For one of the alternative questions of the questionnaire—“How much do you think is reasonable for organic food being more expensive than ordinary food?”—85% of the respondents chose less than 30%, which may reflect the premium of organic food recognized by most residents, and is instructive to the government’s promotion policies and manufacturers’ supply strategies.

### 5.2. The Role of Government Regulation in Organic Food Consumption

Although the theory of planned behavior is a mature method for studying food consumption, a lot of key information has been missed when explaining willingness and behavior merely with traditional attitudes, subjective norms, and perceived behavior control. External intervention, except for external public opinion, cannot be included in the analysis, as it lacks a guiding value for how it influences individual willingness. Ma et al. [25] and Metcalf et al. [44] added external intervention factors as mediators or moderator variables to the model to enhance the explanatory power, similar to the idea in this paper.

In China, government regulation is an external intervention factor that cannot be ignored in almost all social and economic activities [26]. However, existing research on food consumption often takes government intervention as an endogenous variable and does not analyze it separately, which may lead to deviations in the estimation of path coefficients influenced by various psychological factors. Su et al. [24] mostly focus on the role of the government in sales links such as food certification and sales supervision, indicating that the government’s credibility commitment promotes the transformation of consumer’s willingness into behavior. This conclusion is also confirmed in this study, but these studies only focus on some aspects of the government’s regulation. In this paper, the government’s work in the entire chain of food production and sales is divided into two types of regulatory roles, and the TPB model is improved so as to explain the psychological impact of government regulation on the consumption of organic food by Chinese urban residents. After the introduction of the moderator variable of the government, the coefficient of determination of consumption willingness and consumption behavior reached more than 60% and 40%, respectively. Compared with Bilbiie et al.’s food consumption research [52] conducted via the traditional theory of planned behavior, which provides less than 40% of the explanatory power [51], the improvement in the model is verified to be reasonable and practical.

(1)The role of government food production support work

It is the innovation of this paper to take the government’s food production support work as the moderator variable of the influence path of “food safety concern on organic food consumption willingness”, which directly links residents’ perception of agricultural production and that of food safety, and then extends to organic food consumption willingness. In China, the government is deeply involved in agricultural production through agricultural production subsidies and agricultural benefit policies, as well as direct investment in rural production and living environments, agricultural science and technology research and development, and pollution control. The work directly affects urban residents’ perceptions of food. Although Su et al.’s research [24] also found this phenomenon, it did not consider the possible positive and negative impacts of the government’s food production support work. The simultaneous verification of hypotheses H2a and H2b may illustrate two aspects: first, consumers’ concerns about agricultural production safety are an important cause of food safety concern, and government regulation is conducive to alleviating such concerns. Second, although the government’s support for agricultural production will weaken the driving force of “food safety concern on the demand for organic food”, it will make up for the driving force on other psychological paths. For example, residents will be more confident that the government has the ability to supervise the sources of food production, and that the environmental, technical and human conditions for the production of organic food can be provided, which generally promotes the demand for organic food. After all, it is not the right approach to rely on residents’ food safety concerns to promote organic food consumption. In addition, although the results of H2c are not significant, it may indicate that the influence of the government’s food production support work on residents’ net benefit perceptions of organic food is not clear. Some respondents may think that “the government supports agricultural production well, and the quality of ordinary agricultural products should be good, so there is no need to buy organic food at high prices”, while others may think that “the government supports agricultural production well, and the quality of organic food should be better”. This once again reflects that Chinese residents have doubts about whether the quality and benefits of organic food are better than those of ordinary food.

(2)The role of the government’s food consumption guarantee work

Consumption willingness does not always convert into consumption behavior. Xu et al. [53] believe that the important reason is that due to asymmetric information, monopoly, and large enterprises in food production and sales, consumers are often a vulnerable group in food trading, and can only distinguish food quality based on publicity and experience. It is difficult and costly to protect rights when encountering fraud. This kind of concern is the key obstacle to converting willingness into behavior. As Gong et al. [47] once proposed, the government needs to provide authoritative certification for food quality to reduce the information cost of consumers and regulate food processing and transportation to reduce unqualified products. He and Shi [49] also recommend improving the legal systems of food safety and guiding public opinion to create a good market environment, etc. The positive effects of the government’s food consumption guarantee work on consumers’ willingness and behavior to consume organic food are comprehensively verified by H2d and H2e in this study.

(3)Comparison of respondents’ satisfaction with various government regulations

Figure 5 shows the evaluation results of respondents’ satisfaction with the government’s supervision in the entire chain. The respondents are dissatisfied with the government’s work in the supervision of livestock and poultry drugs (GP1), that of fertilizers and pesticides (GP2), and the regulation of outer packaging (GC2), food processing (GC4), and food charity (GC7). These are the key links and regulatory difficulties of food safety, and are the guidelines for improving government work in the future. Respondents are highly satisfied with the government’s work in improving the rural environment (GP4), supporting agricultural science and technology (GP5), perfecting laws and regulations (GC5), and strengthening food safety publicity (GC6), which are also key areas that China has focused on in recent years. In particular, after years of governance, respondents’ satisfaction with the government regulation of food safety certification and traceability (GC1) has exceeded 50%. In conclusion, the respondents’ satisfaction with the government’s efforts to improve the agricultural and rural environment, support agricultural science and technology, and support farmers made up for their dissatisfaction with the supervision of livestock and poultry drugs, pesticides, and fertilizers, making the overall moderating effect of GP positive. The role of strict laws and credible certification in enhancing consumer confidence enables the government’s food consumption guarantee work (GC) to promote the conversion of residents’ willingness to consume organic food into behavior.

### 5.3. The Influence of Respondents’ Personal Factors on Organic Food Consumption

This study proves that an increase in income will promote the willingness to consume organic food. This finding is similar to that in Zámková et al.’s study [51]. With reference to the study of Fish et al. [9], this may indicate that high-income people have both the ability to purchase high-end food and stronger health demand. Although the number of elderly family members and children has a positive impact on the willingness to consume organic food, the value is small (β = 0.007). The influence of chronic diseases suffered by individuals and family members on the consumption willingness of organic food (β = 0.073) is even higher than that of attitude (β = 0.065), reflecting that the health concepts of Chinese residents are still relatively backward; that is, the illness can stimulate the consumption desire of organic food, and the attention paid to ordinary health maintenance is insufficient. According to the answers to the alternative questions in this study (Figure 3), most of the respondents agree that organic food tastes better and can reduce diseases, but are noncommittal regarding whether eating organic food can prolong life, or make families happy and feel honored. More than 30% of the answers to each question are “I don’t know”, reflecting that the respondents have an unclear understanding of the effects and purpose of eating organic food, which once again illustrates the importance of scientific publicity.

### 5.4. Expanding the Connotations of “Organic food Consumption”

It is believed in this study that research on food consumption should not only be limited to food, but also focus on the environment and conditions associated with food production and consumption, and whether consumers are willing to make efforts to improve it. In fact, some scholars have already made similar attempts. Suzuki et al. [54] believed that people who are willing to consume organic food were also more inclined to pay for ecotourism and education to provide support for the production and environmental protection of organic food; Pirani et al. [55] believed that the safety-oriented food consumption behavior of the Gulf countries was closely related to agricultural production and the development of tourism in these countries.

Accordingly, in this paper, the measurement of willingness and behavior has also been improved. Traveling to rural areas and participating in the social supervision of food safety are also included in the investigation of consumption willingness and behavior, and have passed the test of confirmatory factor analysis. The logic behind this is that the production, sales and promotion of organic food also contribute to the environment and the social awareness of environmental protection. Tourism consumption in rural ecological environment, the direct purchase of organic food, and participation in the cause of food safety should also be included in the “broad sense” of organic food consumption. The questionnaire results offer some new findings: compared with Manuela Vega Zamora et al.’s finding [56] that self-interest motivation is the main factor to promote the consumption of organic food, in this study, most of the respondents who are willing to buy organic food are also willing to participate in the promotion of organic food and food safety supervision, which may indicate that the altruistic motivation of Chinese residents is taking shape.

## 6. Conclusions

The organic food market in China holds large potential. The food safety concern of Chinese residents is an important factor to promote the formation of organic food consumption willingness. Under the current income and organic food price level, Chinese residents pay the most attention to the comparison between the “economic and energy costs” and the “benefits of alleviating food safety concerns” of purchasing and consuming organic food. The public opinion environment, the discussion on food safety issues, and being persuaded to buy healthy food by relatives, friends and colleagues will also significantly enhance the residents’ concern about food safety and promote the willingness to buy organic food. However, the influence of residents’ own food safety feelings is relatively low. This is also in line with Chinese people’s characteristics of being budget conscious and being concerned about “face-saving”.

Although the government’s support and supervision of food production will weaken the driving effect of food safety concern and benefit perception on organic food consumption willingness, it will still promote organic food consumption willingness on the whole. The government’s supervision of food processing, transportation, sales and packaging, as well as laws and regulations, publicity, and support for public welfare undertakings of food safety, encourages the conversion of organic food consumption willingness into behavior.

## 7. Managerial Implications

Consumer demand driven by food insecurity is fragile and unhealthy. After all, the difference between “being forced” and “actively loving” is huge. Relevant promotion strategies can start from the scientific publicity of the effects of organic food and the guidance of the public opinion environment. For example, scholars in different fields can be invited to explain the possible effects of different types of organic food on improving nutrition and health. The display of the production process and the technology of organic food is also conducive to directly describing the good production environment to consumers, forming psychological hints of healthy nutrition, and explaining the rationality of the price premium of organic food. Residents may gradually believe through such promotion activities that organic food is worth the money.

On the other hand, the government’s support for the construction of an agricultural production environment and green agricultural technologies can enhance residents’ trust that “organic food is genuine”. The government can also certify villages with an excellent ecological environment, encourage urban residents to travel and consume organic food, and increase the opportunities to understand and buy organic food. It is also necessary to continue to improve the protection of consumers’ rights and interests, reduce information transaction costs and avoid adverse selection by means of clear identification, strict evaluation and inspection standards, the use of new digital technologies to provide organic food traceability channels, etc. The positive guidance that “agricultural production can ensure the quality of organic food” may gradually stimulate the spontaneous demand of residents for organic food and replace the demand caused by food insecurity.

The government should comprehensively reduce costs in the entire chain of organic food production, including processing, transportation, cold chain logistics, storage and sales by means of taxation, preferential water and electricity prices, technology R&D and application, reasonable origin market layout, the acceleration of administrative approval, and green channels for fast transportation. Enterprises should also improve their operations, develop digital consumption, open up direct sales channels, and lower intermediate costs, as well as support experiential consumption and customized consumption, and help residents understand the production process and value of organic food. Controlling the premium of organic food within a reasonable range will achieve a win–win situation.

## 8. Limitations

First, since most sales of organic food in China are in large supermarkets and shopping malls, the spatial distribution of the respondents was considered during the offline questionnaire distribution of this study. However, for the online questionnaire, it was difficult to identify the specific locations of the respondents in Beijing, which may slightly interfere with the randomness of the sampling. This kind of error is acceptable in most cases [36].

Second, due to the limitation of space, separate studies were not conducted for identifying the role of government work in the entire chain of food production and sales supervision. The individual moderating roles of these different areas deserve further exploration.

## Figures and Tables

**Figure 1 foods-11-02965-f001:**
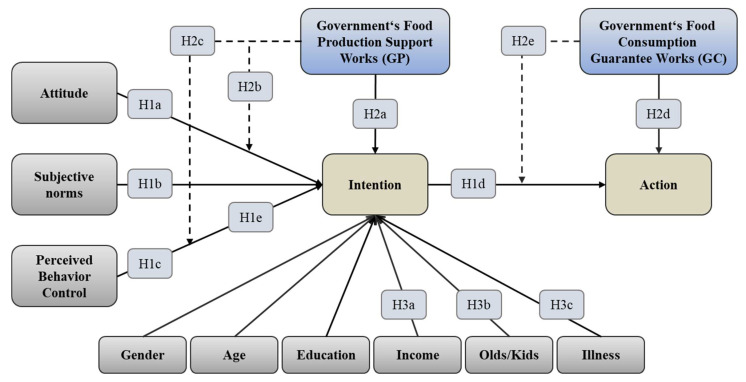
Schematic diagram of SEM model with the moderating effect of satisfaction with government regulation.

**Figure 2 foods-11-02965-f002:**
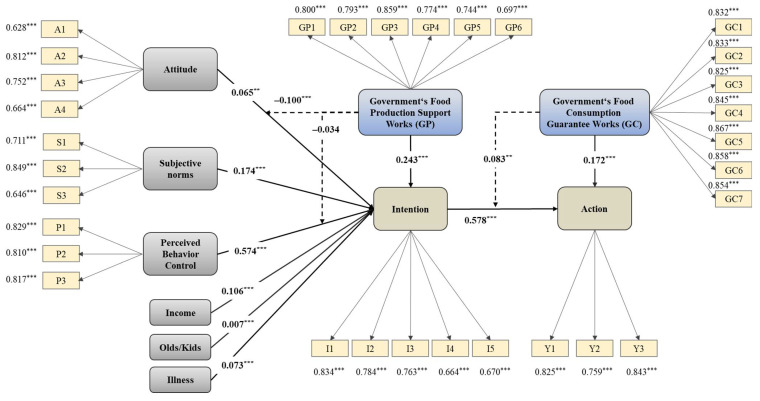
Structural model with path coefficients and levels of significance. *** *p* < 0.01. ** *p* < 0.05.

**Figure 3 foods-11-02965-f003:**
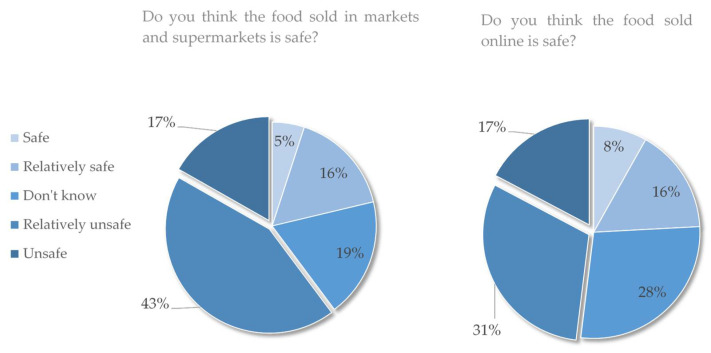
Results of the 799 respondents’ sense of food safety in offline and online sales.

**Figure 4 foods-11-02965-f004:**
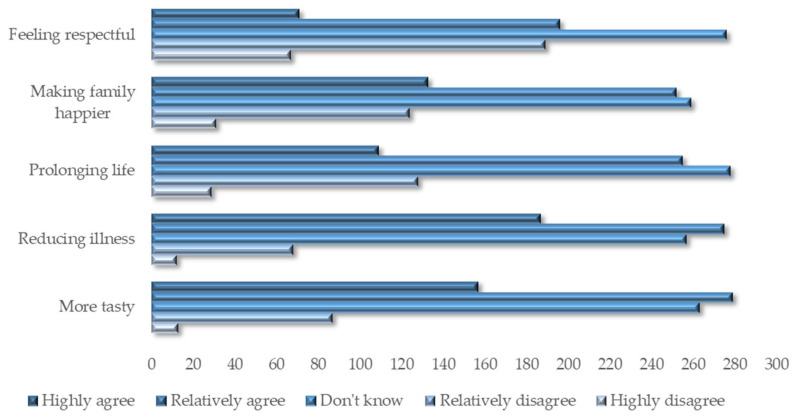
Respondents’ perception of various benefits of organic food.

**Figure 5 foods-11-02965-f005:**
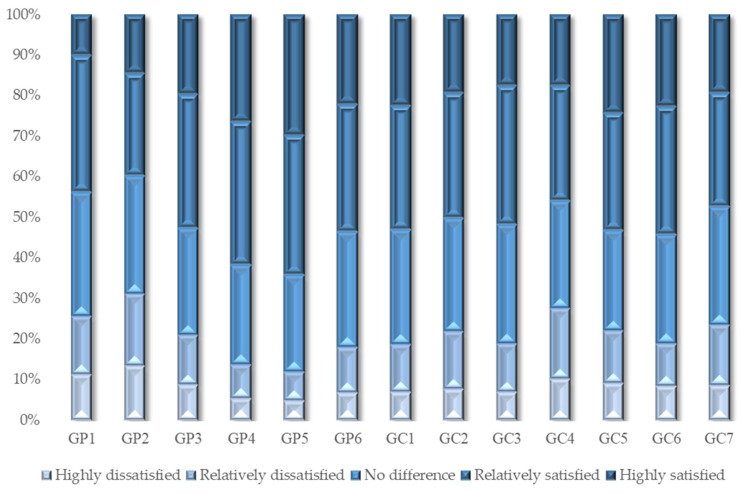
Results of satisfaction with government regulation.

**Table 1 foods-11-02965-t001:** Descriptive statistics.

Individual Characteristics	Participants (%)
n	799
Gender	
Male	297 (37.17%)
Female	502 (62.83%)
Academic qualifications	
High school or below	99 (12.39%)
Associate degree	137 (17.15%)
Undergraduate	378 (47.31%)
Master’s degree or above	185 (23.15%)
Age	
19–30 years old	198 (24.78%)
31–45 years old	363 (45.44%)
46–59 years old	218 (27.28)
≥60 years old	20 (2.5%)
Average age	38.36
Family income	
<CNY 4000	54 (6.7%)
CNY 4000–10,000	225 (28.16%)
CNY 10,000–20,000	282 (35.29%)
CNY 20,000–50,000	192 (24.03%)
>CNY 50,000	46 (5.76%)
Number of the elderly and children	
0	108 (13.52%)
1	190 (23.78%)
2–3	278 (34.79%)
4	156 (19.52%)
≥5	67 (8.39%)
Number of chronic diseases suffered by oneself and family members	
0	33 (4.13%)
1	73 (9.14%)
2	169 (21.15%)
3	277 (34.67%)
≥4	247 (30.91%)

**Table 2 foods-11-02965-t002:** Construct measures, items, loadings, reliability and validity.

	Item	Loading	Composite Reliability (CR)	Average Variance Extracted (AVE)	Cronbach’s Alpha
Attitude	A1	0.628 ***	0.808	0.515	0.792
	A2	0.812 ***			
	A3	0.752 ***			
	A4	0.664 ***			
Subjective norms	S1	0.711 ***	0.782	0.548	0.700
	S2	0.849 ***			
	S3	0.646 ***			
Perceived behavioral control	P1	0.829 ***	0.859	0.670	0.859
	P2	0.81 ***			
	P3	0.817 ***			
Intention	I1	0.834 ***	0.861	0.556	0.830
	I2	0.784 ***			
	I3	0.763 ***			
	I4	0.664 ***			
	I5	0.670 ***			
Action	Y1	0.825 ***	0.851	0.656	0.853
	Y2	0.759 ***			
	Y3	0.843 ***			
GP	GP1	0.800 ***	0.902	0.608	0.900
	GP2	0.793 ***			
	GP3	0.859 ***			
	GP4	0.774 ***			
	GP5	0.744 ***			
	GP6	0.697 ***			
GC	GC1	0.832 ***	0.946	0.714	0.946
	GC2	0.833 ***			
	GC3	0.825 ***			
	GC4	0.845 ***			
	GC5	0.867 ***			
	GC6	0.858 ***			
	GC7	0.854 ***			

*** *p* < 0.01. The effects of gender, age, education level and income, the number of elderly and children in the family, and the types of chronic diseases suffered by oneself and family members on organic food consumption willingness and behavior were controlled.

**Table 3 foods-11-02965-t003:** Fornell–Larcker criterion test.

	Attitude	Subjective Norms	Perceived Behavior Control	Intention	Action	GP	GC
Attitude	0.718						
Subjective norms	0.225	0.74					
Perceived behavior control	0.37	0.351	0.818				
Intention	0.392	0.426	0.61	0.746			
Action	0.183	0.198	0.284	0.465	0.809		
GP	−0.297	−0.238	−0.104	−0.009	−0.003	0.779	
GC	−0.283	−0.194	−0.169	−0.017	−0.006	0.686	0.845

**Table 4 foods-11-02965-t004:** Collinearity statistics (variance inflation factor (VIF)).

	Intention	Action
Attitude	1.34	
Subjective norms	1.31	
Perceived behavior control	1.73	
Intention		1.31
Action		
CP	2.00	
GC		2.03
Attitude × GP	1.84	
Perceived behavior control × GP	1.39	
Intention × GC		1.61

**Table 5 foods-11-02965-t005:** Path influence coefficient and significance.

			Path Coefficients
Intention	←	Attitude	0.065 **
Intention	←	Subjective norms	0.174 ***
Intention	←	Perceived behavioral control	0.574 ***
Intention	←	GP	0.243 ***
Intention	←	Attitude × GP	−0.100 ***
Intention	←	Perceived behavioral control × GP	−0.034
Action	←	GC	0.172 ***
Action	←	Intention	0.578 ***
Action	←	Intention×GC	0.083 **
Intention	←	Income	0.106 ***
Intention	←	Illness	0.073 ***
Intention	←	Olds/kids	0.007 ***
Intention	←	Gender	0.052
Intention	←	Age	0.101
Intention	←	Education	0.033

*** *p* < 0.01. ** *p* < 0.05. ” ←” represents the direction of influence path.

**Table 6 foods-11-02965-t006:** Confirmed hypotheses.

Hypothesis	Description	Supported	Rejected
H1a	Awareness and concern about food safety will increase the willingness to consume organic food.	√	
H1b	The public opinion environment of food safety and the persuasion of others will enhance the willingness to buy organic food.	√	
H1c	The higher the residents’ sense of net benefits from organic food, the stronger the willingness to consume organic food.	√	
H1d	Chinese residents’ willingness to buy organic food after considering various factors will form the consumption behavior of organic food.	√	
H1e	Considering the “cost–benefit” factor has a greater impact on the willingness to buy organic food than other factors.	√	
H2a	The higher the residents’ satisfaction with the government’s food production support work, the higher their willingness to buy organic food.	√	
H2b	Residents’ satisfaction with government’s food production support work may inhibit the promotion of food safety concerns related to organic food consumption willingness.	√	
H2c	Residents’ satisfaction with the government’s food production support work may weaken the driving effect of the sense of the net benefit of organic food on the willingness to consume organic food.		×
H2d	Satisfaction with the government’s food consumption guarantee work will promote consumers to buy organic food.	√	
H2e	Satisfaction with the government’s food consumption guarantee work will prompt consumers to turn their willingness to buy organic food into behavior.	√	
H3a	The higher the income of residents, the stronger their willingness to buy organic food.	√	
H3b	Having more elderly family members and children in the family will increase the willingness to buy organic food.	√	
H3c	The more individuals and family members that suffer from serious chronic diseases, the stronger the willingness to buy organic food.	√	

“√” or “×” means the hypothesis is supported or rejected in this research.

## Data Availability

The data that support the findings of this study are available from the corresponding authors upon reasonable request.

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
