# Peer review of "Influence of Food Safety Concerns and Satisfaction with Government Regulation on Organic Food Consumption of Chinese Urban Residents"

_foods, 2022, doi:10.3390/foods11192965_

Round 1

Reviewer 1 Report

Dear authors, I suggest the abstract start with setting the objective, then indicating the target sample of respondents, discussing the statistical methods used, pointing out the most important results and finally presenting some conclusions based on the results from your study.
The article has a theoretical background section that is quite elaborate and an Introduction section that is too long. The Introduction sections should be shortened to just indicate why your topic is important and worth investigating. The Introduction section should end with an indication of the purpose of your manuscript.
Does the research sample in any way reflect the population composition of the City of Beijing?
Discussion, this is the section where authors confront their results obtained during the study with those of other authors. That's what I don't have here. In the discussion, the authors refer literally to several items. This should be reworded.
To enrich the discussion, I suggest referring to the works of the following authors on consumer behavior in the market of organic products: Zámková, M., Guiné, R.P.F., Śmiglak-Krajewska, M., Teixeira, S.F., Vega-Muñoz, A.
Also, I don't see a Conclusion section in which the authors summarize in points the most important achievements of their work.

Author Response

Thanks for your kindly comments.

1Dear authors, I suggest the abstract start with setting the objective, then indicating the target sample of respondents, discussing the statistical methods used, pointing out the most important results and finally presenting some conclusions based on the results from your study.

Thanks for your valuable comments! We have revised the writing logic and content of the abstract accordingly, and shortened the length of the abstract.

Please refer to the front page of the paper.

2The article has a theoretical background section that is quite elaborate and an Introduction section that is too long. The Introduction sections should be shortened to just indicate why your topic is important and worth investigating. The Introduction section should end with an indication of the purpose of your manuscript.

Thank you for the kind advices. As suggested, we have simplified the introduction, clarified the importance of the study, and indicated the purpose of the paper. Moreover, we have also reduced the length of the theoretical background part, adjusted the writing logic and merged some paragraphs with similar contents.

Please see lines 51-418 for detail.

3Does the research sample in any way reflect the population composition of the City of Beijing?

Thank you for pointing it out. The relationship between the sample structure and the population structure of Beijing is explained as follows:

First of all, the purpose of this study is to understand how food safety concerns and government regulation affect residents' willingness to consume organic food. The popularity of organic food in China is low, so the survey should have specific target groups. If the interviewee has not heard of organic food, or does not buy vegetables and cook, he is not a participant in the organic food market, and this sample has no reference value. Therefore, we designed two "screening questions" in the questionnaire to screen the target groups, including "do you know and have seen organic food" and "do you usually buy vegetables and cook". The target group structure selected by this method may be different from the total population structure of Beijing. Referring to [Shen, C and et.al.. A Study on the Influences of Fiscal and Tax Policies on Low-carbon Consumption Behavior of Residents. Taxation Research 2016, 2, 98-104.] and [Larmarange, J and et.al.. Feasibility and Representativeness of a Random Sample Mobile Phone Survey in Cote D'ivoire. Population 2015 71, 123-134.], There are also lots of the same methods as we have used.

Besides, in the study, we can only compare gender, age and income obtained in the survey with the available official data of Beijing because of the absence of authoritative and detailed data on population segmentation structure:

According to the data of the Beijing Municipal Bureau of statistics, from the perspective of income level, the average monthly income of households in Beijing in 2021 is 18140 yuan, and the median is 13050 yuan. The average income of our sample is 19230 yuan, with a median of 14160 yuan, which are close to the official statistics.

According to the seventh national census, the average age of Beijing's population is 39.2 years old. Since our target group does not include children and the elderly, the average age of the sample is 38.36 years old, which is close to the official statistics.

As for gender, the ratio of men to women in Beijing is 51:49; In this study, the ratio is about 37:62, which has some differences; This is because, in Chinese, they are women who buy food and cook most in families. Thus, such difference is also reasonable.

To sum up, we believe that the income and age of the sample do not deviate from the population situation of Beijing, and the difference in gender structure is reasonable. The representativeness of the samples is acceptable.

  1. Discussion, this is the section where authors confront their results obtained during the study with those of other authors. That's what I don't have here. In the discussion, the authors refer literally to several items. This should be reworded.

According to your comments, we have rewritten the discussion part. We focus on comparing our major findings with the other relevant studies, highlighting the new findings of this paper.

Please see details in section “Discussion”.

  1. To enrich the discussion, I suggest referring to the works of the following authors on consumer behavior in the market of organic products: Zámková, M., Guiné, R.P.F., Śmiglak-Krajewska, M., Teixeira, S.F., Vega-Muñoz, A.

Thank you for the detailed advices and sharing these highly relevant works. We’ve added and referred these articles.

  1. Also, I don't see a Conclusion section in which the authors summarize in points the most important achievements of their work.

Thank you for your kind advices. As suggested, we’ve added the "conclusion" part to summarize the valuable findings of our paper. Moreover, we’ve also addressed the major achievement of this work and its implications, as well as limitation.

This writing style also refers to articles published by foods, such as Zheng, G. - W; Akter, N.; Siddik, A.B.; Masukujjaman, M. Organic Foods Purchase Behavior among Generation Y of Bangladesh: The Moderation Effect of Trust and Price Consciousness. Foods 2021, 10, 2278. https://doi.org/10.3390/foods10102278

Reviewer 2 Report

I must say that the manuscript is well written with comprehensive information. However, the manuscript itself is VERY long to read and I'd encourage the authors to merge and prioritise some section from Section 1 & 2.

Did the authors check for internal reliability and robustness of their questionnaire? Cronbach alpha for example? OK it is measured but not mentioned until later in the results section, please add this to the procedure to cover ALL the statistical methods that was used.

Was there any ethics approval for this study in data collection? If yes, please state.

The authors stated that <300s responses are omitted, what was the ideal time of completing the survey? And how is this limit reached?

Why df of 17 and p<.01? Why are these parameters selected?

Using z-scores are an indicator of outliers isn't the most suitable technique, why did the author use this approach?

Figure 1, Intention rather than intension?

I reckon the author should subsection their Procedure section clearly so that it is clear to the readers what was exactly done and to what purpose

Table 2 Income, elderly, etc. is it supposed to be there?

It is common approach to use 5% rather than 10% alpha level, why is 10% investigated here?

Discussion do attempt to explain their results, but again I reckon the authors might benefit by subheading their results to show what is being discussed, the topic jumped quite often and is hard to follow.

Author Response

Thanks for your kindly comments.

  1. I must say that the manuscript is well written with comprehensive information. However, the manuscript itself is VERY long to read and I'd encourage the authors to merge and prioritise some section from Section 1 & 2.

Thank you for the valuable suggestion. Reviewer 1 also provides similar comments. As suggested, we’ve shorten the introduction, clarified the importance of the study, and indicated the purpose and marginal contribution of the paper.

Please find detail in section “introduction”.

  1. Did the authors check for internal reliability and robustness of their questionnaire? Cronbach alpha for example? OK it is measured but not mentioned until later in the results section, please add this to the procedure to cover ALL the statistical methods that was used.

Thank you for your kind reminder and suggestion. We have added statistical methods and indicators that were used in the “procedure” section, including Cronbach alpha, CR, AVE, Chi-square/df GFI, AGFI, and RMSEA.

Please see line 558~568 for detail.

  1. Was there any ethics approval for this study in data collection? If yes, please state.

Thank you for pointing it out. We all agree that ethics approval is very important. Before we begin to conduct the survey, we’ve checked and consulted with the ethics department in our universities. According to the relevant regulations of China, this study is not within the scope of ethical regulation. Since questions are about the food consumption behavior of residents which do not cause additional physiological burden to the respondents, nor will it cause additional expenses to the respondents.

In addition, the questionnaire is completely anonymous and the information of the respondents is strictly confidential. This has been stated in the introduction of our questionnaire, and the online platform "questionnaire star" that launched the questionnaire also has relevant policies.

For details, please refer to https://www.wjx.cn/wjx/license.aspx?type=1&ivk_sa=1024320u.

Therefore, our study did not carry out ethics approval.

  1. The authors stated that <300s responses are omitted, what was the ideal time of completing the survey? And how is this limit reached?

Thank you for asking the cutting-off point. We should address it more in our previous manuscript.

The length of filling time is a meaningful indicator for screening whether the respondent answers the questions carefully. The ideal time of filling in the questionnaire is obtained through pre-testing. As we said in the article, before the questionnaire was distributed, we invited 25 volunteers to fill in the questionnaire in a quiet and comfortable setting. We confirm that all the pilot invited to fill in the questionnaire is carefully, and the shortest filling time is about 300 (±5, consider timing error) seconds. Therefore, 300-seconds has been used for as a limit.

We’ve explained the reasons for this in the article.

  1. Why df of 17 and p<.01? Why are these parameters selected?

Thank you for the question.

1) The degree of freedom is determined by the number of distinct sample moments and distinct parameters to be estimated in the SEM model. The standard of p <.01 was selected in this manuscript based on strict premise of Mahalanobis distance test, referring to [Metcalf, D.A.; Wiener, K.K.; Saliba, A.; Sugden, N Evaluating the acceptance of hemp food in Australian adults using the Theory of Planned Behavior and Structural Equation Modelling. Foods 2021, 10, 2071.].

2) The variable selection is based on the planned behavior theory and relevant empirical studies. We’ve added more explanation in the “Theoretical Background” and“Questionnaire Design”sections. Selected variables reflect different aspects including residents' food safety concerns, government job satisfaction, and organic food consumption willingness and behavior.

  1. Using z-scores are an indicator of outliers isn't the most suitable technique, why did the author use this approach?

Thank you for the kind question!We’ve provided more explanation in our revised manuscript. Z-score describes the distance between a given measured value x and the average value. It is generally considered that if Z-score > 1.96, there may be outliers.  In questionnaire research, Z-score test is also a commonly used method, referring to [Metcalf, D.A.; Wiener, K.K.; Saliba, A.; Sugden, N. evaluating the acceptance of hemp food in Australian adults using the theory of planned behavior and structural equation modeling. Foods 2021, 10, 2071.] and other studies.

In addition, we agree that Z-score is not the best method for outlier test. Therefore, Mahalanobis distance test is used to filter outliers before Z-score in this manuscript, and Z-score is a supplementary test for the filtered sample points to ensure that outliers are completely eliminated.

  1. Figure 1, Intention rather than intension?

Sorry for the typo and thank you for the careful check! We’ve corrected the error in the figure and checked the full article.

  1. I reckon the author should subsection their Procedure section clearly so that it is clear to the readers what was exactly done and to what purpose

Thank you for your kind advices! We found this comment is very helpful.

To improve readability, three subsection headings have been added into “Procedure” section, named “Data Collection” “Samples” “Model Analysis”. It makes “Procedure” section much clearer.

Please see line 534~590 for detail.

  1. Table 2 Income, elderly, etc. is it supposed to be there?

We agree. As suggested, we have delete these items and added a note bellow table 2 for better explaination.

Please see line 624~647 for detail. (Table 2)

  1. It is common approach to use 5% rather than 10% alpha level, why is 10% investigated here?

Thank you for the detailed comments!

This paper is a research paper of economic management. Generally speaking, in the economic and management discipline, the commonly accepted alpha levels are 10%. Therefore, 10% alpha level is investigated.

Also, please note that although we listed 10% alpha level, the significance level of all the results in this article is above 5%.

For similar practices, please see: [Rodriguez-Bermudez, R. and et al.. Consumers' perception of and attitudes towards organic food in Galicia (Northern Spain). International Journal of Consumer Studies 2020, 44, 206-219.] and [Bîlbîie, A.; Druică, E.; Dumitrescu, R.; Aducovschi, D.; Sakizlian, R.; Sakizlian, M. Determinants of Fast-Food Consumption in Romania: An Application of the Theory of Planned Behavior. Foods 202110, 1877.].

  1. Discussion do attempt to explain their results, but again I reckon the authors might benefit by subheading their results to show what is being discussed, the topic jumped quite often and is hard to follow.

We found this comment is helpful. According to your suggestions, we’ve added subtitles in the discussion part, sorted out the logical order of the exposition, highlighted the new findings of this paper, and compared these findings with the latest research results of other scholars.

Round 2

Reviewer 1 Report

The article has been corrected as indicated. It is acceptable as it stands. I have no more comments.

Good luck

Author Response

Dear reviewer,

Thanks for your comments. 

Wish you a good day.

Reviewer 2 Report

Comments have been addressed thanks. Just minor grammatical mistakes, data collection rather than data collecting in Section 3.3

It is unclear to me whether the Figures in Discussion section is a part of this study or other's study, if it's others then it's fine (check for copyright), but if it is this study then shuffle it to results section

Author Response

Dear reviewer,

Thanks for your comments.

Response

Reply to the comments and suggestions of second reviewer:

1Comments have been addressed thanks. Just minor grammatical mistakes, data collection rather than data collecting in Section 3.3.

Thanks for your valuable comments! The mistakes have been corrected.

2The article has a theoretical background section that is quite elaborate and an Introduction section that is too long. The Introduction sections should be shortened to just indicate why your topic is important and worth investigating. The Introduction section should end with an indication of the purpose of your manuscript.

Thank you for your kind advices. As suggested, we’ve moved the figure of the discussion part forward to the result part, and adjusted the corresponding description and discussion.

Please see “Result” section for detail.

Wish you a good day.